# Biocompatibility Analysis of Bio-Based and Synthetic Silica Nanoparticles during Early Zebrafish Development

**DOI:** 10.3390/ijms25105530

**Published:** 2024-05-18

**Authors:** Cinzia Bragato, Roberta Mazzotta, Andrea Persico, Rossella Bengalli, Mariana Ornelas, Filipa Gomes, Patrizia Bonfanti, Paride Mantecca

**Affiliations:** 1POLARIS Research Center, Department of Earth and Environmental Sciences, University of Milano-Bicocca, 20126 Milan, Italy; r.mazzotta2@campus.unimib.it (R.M.); a.persico@campus.unimib.it (A.P.); rossella.bengalli@unimib.it (R.B.); patrizia.bonfanti@unimib.it (P.B.); paride.mantecca@unimib.it (P.M.); 2CeNTI—Centre for Nanotechnology and Smart Materials, Rua Fernando Mesquita 2785, 4760-034 Braga, Portugal; mornelas@centi.pt (M.O.); fgomes@centi.pt (F.G.)

**Keywords:** SiO_2_-NPs, zebrafish embryos, neurotoxicity, inflammation, macrophages

## Abstract

During the twenty-first century, engineered nanomaterials (ENMs) have attracted rising interest, globally revolutionizing all industrial sectors. The expanding world population and the implementation of new global policies are increasingly pushing society toward a bioeconomy, focused on fostering the adoption of bio-based nanomaterials that are functional, cost-effective, and potentially secure to be implied in different areas, the medical field included. This research was focused on silica nanoparticles (SiO_2_-NPs) of bio-based and synthetic origin. SiO_2_-NPs are composed of silicon dioxide, the most abundant compound on Earth. Due to their characteristics and biocompatibility, they are widely used in many applications, including the food industry, synthetic processes, medical diagnosis, and drug delivery. Using zebrafish embryos as in vivo models, we evaluated the effects of amorphous silica bio-based NPs from rice husk (SiO_2_-RHSK NPs) compared to commercial hydrophilic fumed silica NPs (SiO_2_-Aerosil200). We evaluated the outcomes of embryo exposure to both nanoparticles (NPs) at the histochemical and molecular levels to assess their safety profile, including developmental toxicity, neurotoxicity, and pro-inflammatory potential. The results showed differences between the two silica NPs, highlighting that bio-based SiO_2_-RHSK NPs do not significantly affect neutrophils, macrophages, or other innate immune system cells.

## 1. Introduction

According to Commission Recommendation 2022/C 229/01, nanomaterials (NMs) are natural, incidental, or manufactured materials containing solid particles present in an unbound state, as aggregates or agglomerates. They must meet certain criteria in the number size distribution, and present one or more dimensions between 1 and 100 nm.

The first scientist who thought about the manipulation of materials at the atomic scale was Richard Feynman, a physicist who is still considered the father of nanotechnology [1]. This term refers to the science, engineering, and technology conducted at the nanoscale (1–100 nm), in which unique phenomena enable new applications in a wide range of fields including chemistry, engineering and electronics, physics, biology, and medicine (NNI—National Nanotechnology Initiative, https://www.nano.gov/, accessed on 2 February 2023). To date, nanotechnology-based drug-delivery platforms, diagnostic and sensing, have gained increasing attention in the last years since nanomaterials may offer many advantages, such as tailored drug delivery, reduced side effects, and greater stability of formulations [2,3]. The field of nanomedicine has progressed rapidly, and the new frontier is to use bio-nanomaterials, potentially unharmful to humans and the environment.

Among the nanoparticulate systems, silica nanoparticles (SiO_2_-NPs) are recognized for their powerful properties within biological applications. Inorganic SiO_2_-NPs are characterized by high surface area, biocompatibility, and tissue stability under acid conditions [4].

Silica dioxide, more commonly named silica, is the most abundant compound in the Earth’s crust, where it can be found as silicate minerals, as well as in plants, cereals in particular [5]. Different types of SiO_2_-NPs exist and can be manipulated to have specific characteristics to be exploited for precise use [6,7,8,9,10,11].

Currently, silica NPs are finding wide applications in the biomedical field as drug carriers [12,13,14], diagnostic/imaging tools [15], and molecular detection/targeted therapy [16]. For this reason, the investigation of the neurotoxic and inflammatory effects at the molecular level is mandatory.

In this research, the effects of two different silica NPs were compared. The first SiO_2_-NP is commercial AEROSIL^®^ 200 (SiO_2_-Aerosil200 NP), a hydrophilic fumed silica with a specific 200 m^2^/g surface area. This NP is employed in paints and coatings, unsaturated polyester resins, HCR- and RTV 2K silicone rubber, adhesives and sealants, printing inks, cable compounds and gels, plant protection, and food and cosmetics. The second is a newly developed bio-based silica NP derived from rice husk residues (SiO_2_-RHSK NP).

The interest behind the bio-based NPs and nanomaterials (NMs) is related to the fact that these can combine nanotechnology with the typical advantages of renewables, such as biodegradability, biocompatibility, abundance, and low production costs [17,18].

The effects of both NPs were assessed on zebrafish embryos, which offer a valid alternative to mammals for the evaluation of nanomaterials, including the advantage of a whole organism represented by a simplified development compared to higher vertebrates [19,20,21]. Embryos are characterized by many benefits, among them the high level of genetic homology to humans (70%). These characteristics make zebrafish a powerful model system for studying different materials’ toxic effects and for predicting what potentially could happen to human beings. They possess comparable physiological parameters such as blood composition, organ systems, the presence of an innate immune system during early development, and a blood–brain barrier [22].

Moreover, the embryos are valuable for evaluating the epidermal uptake of NPs [23].

In this study, both the silica NPs were characterized by the use of Transmission Electron Microscopy (TEM), Dynamic Light Scattering (DLS), and Fourier Transform Infrared (FTIR-ATR) Spectroscopy, and then evaluated at the toxicological level using the Fish Embryo Acute Toxicity (FET) Test [24]. The results allowed for the selection of the preferable exposure concentration to test the effects of NPs on zebrafish embryos, based on direct contact with the epidermal layer, and the subsequent interaction with different pathways, such as SiO_2_-induced neurotoxicity and innate immune system, evaluated at the molecular and histochemical level.

The results highlighted the more suitable biocompatibility of SiO_2_-RSHK NPs, compared to SiO_2_-Aerosil200, and their safe applicability in the biomedical or other industrial nanotech fields as a new, compelling bio-based material.

## 2. Results

### 2.1. Characterization of Silica Nanoparticles

TEM images showed the morphological characteristics of SiO_2_-Aerosil200 and SiO_2_-RHSK nanoparticles used in the present study. Both NPs exhibited an agglomerated state and an irregular polyhedral shape (Figure 1A). The average size distribution of SiO_2_-Aerosil200 and SiO_2_-RHSK NPs, determined using ImageJ 1.54 (9 February 2023) software, were less than 30 nm for both (Figure 1A). The hydrodynamic diameters and ζ potentials of SiO_2_-Aerosil200 and SiO_2_-RHSK NPs (1 mg/mL) in milliQ water and in fish embryo growing medium (FET) were measured by dynamic light scattering (DLS) and reported in Table 1. The results showed that the hydrodynamic diameter observed in milliQ water and FET is higher for both NPs than the diameter estimated by TEM. This result indicated that both NPs tend to form aggregates. For instance, the elevated standard deviation value measured in the SiO_2_-RHSK NPs hydrodynamic diameter suggested that the agglomerates are not homogeneous, and their measures are between 400 and 700 nm. The agglomeration was also observed in SiO_2_-Aerosil200 NPs, but with homogeneous dimensions, highlighted by the lower standard deviation values, in addition to a low polydispersion index (between 0.08 and 0.7). SiO_2_-Aerosil200 and SiO_2_-RHSK NPs are characterized by negative ζ-potential (−21.03 ± 0.62 mV and −2.80 ± 0.14 mV, respectively). The ζ-potentials measured were ≤−30mV, a threshold value that characterizes a colloidal system considered stable, due to electrostatic repulsions preventing the aggregation of dispersed particles. The SiO_2_-Aerosil200 and SiO_2_-RHSK NPs’ ζ-potential values indicated that repulsive forces are prevalent over attractive forces, presumably referring to the surface charge of the agglomerates and not of the individual NPs, which therefore tend to form agglomerates.

The FTIR-ATR analysis performed on SiO_2_-RHSK and SiO_2_-Aerosil200 NPs showed slight differences between the nanoparticles observed (Appendix A).

### 2.2. SiO_2_-RHSK and SiO_2_-Aerosil200 NPs Do Not Show Acute Toxic Effects or Morphological Defects in Zebrafish Embryos

The FET test was set up using different exposure concentrations of both silica NPs (0.1, 1; 10, and 100 µg/mL). The results did not show embryotoxic and teratogenic effects after SiO_2_-Aerosil200 and SiO_2_-RHSK NP exposure at any concentration (Figure 1B).

The embryos exposed to the highest concentration (100 μg/mL) of both NPs did not show a significant increment in mortality or malformations compared to the control embryos (Ctrl) not exposed. For these reasons, the evaluation of the Lethal Concentration 50 (LC_50_) and the Effective Concentration 50 (EC_50_) was not possible since both have to be assumed to be higher than the highest concentration (100 μg/mL) of the exposure used in our experiments.

Since the effects on mortality rate or hatching delay were negligible for both NPs observed (Figure 1B), the embryos were subjected to morphometric analysis at the end of the exposure period (96 h). The embryos’ length, yolk area, eye distance, eye diameter, and eye area were measured after SiO_2_-Aerosil200 and SiO_2_-RHSK NP exposure at the different concentrations used (0.1, 1, 10, 100 μg/mL) and compared to control embryos not exposed (Ctrl).

The results did not show significant differences between embryos exposed to SiO_2_-Aerosil200 NPs, SiO_2_-RHSK NPs, and Ctrl (Figure 1B).

For instance, the only effects observed on mortality rate and hatching, which were negligible, let us hypothesize that the reason could be due not only to the intrinsic safety that unquestionably characterizes the SiO_2_-Aerosil200 and SiO_2_-RHSK NPs, but also to the fact that these NPs aggregate, being blocked by the chorion (Figure 1C), the natural barrier that protects the embryos during the early developmental stages. This structure can reduce and mitigate the direct effects of NPs on embryo development.

### 2.3. Fluorescent Signal Evaluation in Tg(-3.1neurog1:GFP)sb2 Zebrafish Embryos Is Not Impaired after SiO_2_-RHSK and SiO_2_-Aerosil200 NP Exposure

The *neurogenin1* expression in the neuroectoderm is fundamental for the differentiation of the primary neurons. The zebrafish embryos present distinct sites of primary neurogenesis, represented by the complex pattern of neurogenin1 expression at the neural plate. In the anterior neural plate, distinct groups of cells express this gene. In contrast, the areas of Rohon–Beard sensory neurons, interneurons, and motoneurons are highlighted by three stripes of neurogenin1 expression at the lateral, intermediate, and medial positions of the posterior neural plate. The spatially complex pattern of neurogenin1 expression in the neural plate suggests that this gene is a target of various signals and an integration point of both dorsoventral and anteroposterior positional cues in gastrula and early neurula stage embryos [25].

Given the importance of the expression pattern of this gene, we decided to evaluate SiO_2_-RHSK and SiO_2_-Aerosil200 NPs’ effects in the transgenic zebrafish *Tg(-3.1neurog1:GFP)sb2* [26]. The idea was to observe if SiO_2_-RHSK and SiO_2_-Aerosil200 NPs could impact *neurog1* expression through epidermal exposure (Appendix A).

The *Tg(-3.1neurog1:GFP)sb2* embryos were mechanically dechorionated at the end of the somitogenesis developmental stage (18–20 hpf) and successively exposed to both NPs at a concentration of 100 µg/mL for 96 h. At the end of the exposure period, the GFP signal was evaluated under an epifluorescence stereomicroscope (Figure 2A).

The results did not show significant differences in embryos exposed to SiO_2_-RHSK and SiO_2_-Aerosil200 NPs compared to control embryos not exposed (Figure 2B).

### 2.4. Both SiO_2_-RHSK and SiO_2_-Aerosil200 NPs Did Not Show Neurotoxic Effects at the Molecular Level

To additionally check for possible neurotoxic effects, the embryos at 96 hpf exposed to NPs or not exposed (Ctrl) were processed for RNA extraction. The RNA was subsequently reverse-transcribed to cDNA. For the molecular analyses, we decided to increase the concentration of SiO_2_-Aerosil200 and SiO_2_-RHSK NPs to 200 µg/mL, since Hong and colleagues reported that this concentration significantly increases mRNA expression of immune-related genes [27].

The gene expression levels observed were *β-synuclein (β-syn*), which enables copper ion binding activity, acting upstream or during the differentiation of dopaminergic neurons and on the movement of larvae; *park7*, reported to be involved in the response to oxidative stress and its human ortholog is implicated in the development of Parkinson’s disease, cerebral infarction, and stroke; *rplf13a*, which possesses RNA binding activity and is involved in the negative regulation of transcription; *atg5*, involved in autophagosome assembly; *gfap*, uniquely found in astrocytes in the central nervous system (CNS); ambra1a, involved in the embryonic development of chordates and in the development of skeletal muscle fibers; *ulk1a*, which presents serine/threonine kinase activity and is involved in autophagosome assembly, axon extension, and autophosphorylation; the *uch-1l* gene, characterized by ubiquitin-specific thiol-dependent protease activity and is involved in removing ubiquitin from proteins; *gap43*, which has Calmodulin binding activity and is involved in the regeneration of axons and tissues and plays a pivotal role in axon growth during CNS development; parkin, which has ubiquitin-protein ligase and ubiquitin-conjugating enzyme activity and is involved in several processes, including in the electrons’ transport at the mitochondrial level from NADH to ubiquinone; *pink*1, known to possess serine/threonine kinase activity and involves numerous processes, such as the differentiation of dopaminergic neurons, the development of nucleated erythrocytes, and the regulation of the vascular endothelial growth factor signaling pathway; and *beclin1*, involved in cellular senescence.

The 96 h exposure with the NPs under study, determined a decreasing trend in the expression levels of *β-syn, ambra1a, ulk1a, uch-1l, parkin*, and *pink1*, compared to Ctrl. Since these reductions were not statistically significant, we can conclude that the effects of SiO_2_-RHSK and SiO_2_-Aerosil200 NPs are negligible on these genes’ levels. The same applies to the increased levels of *park7*, *rpfl13a*, *atg5*, and *beclin1*, as not statistically significant compared to Ctrl.

Regarding the *gfap* expression level, we observed an insignificant up-regulation after 200 µg/mL SiO_2_-RHSK NP exposure compared to SiO_2_-Aerosil200 NP-exposed embryos and Ctrl embryos.

To date, the down or up-regulation of the mRNA observed is visible because we used the log2 normalized method to evaluate their expression level. This strategy allows us to considerably appreciate the genes’ up and downregulations, even if not statistically significant [28].

Interestingly, a significantly increased level was observed for the *gap43* gene in embryos exposed to both SiO_2_-RHSK and SiO_2_-Aerosil200 NPs compared to Ctrl (Figure 3). The Gap43, a cytoplasmic protein coded by *gap43*, is expressed in the neuronal growth cones during synaptogenesis and axonal development/regeneration [29].

### 2.5. SiO_2_-Aerosil200 Reduced ccl34a Gene Levels, Related to Macrophages, Compared to SiO_2_-RHSK NPs

The expression levels of genes such as *nf2a*, *nf2b*, *il6*, *ccl34a.4*, *il1β*, *nfkb2*, *nfkbiaa*, *hmox1a*, *cxcl-cic*, *il8*, and *tnfα*, involved in the inflammatory process, were analyzed by qRT-PCR. *Nrf2a* and *nfr2b*, orthologues of the human nuclear factor erythroid 2-related factor 2 (NRF2), are transcription factors that regulate the cellular defense against toxic and oxidative insults [30]. *Tnfα* is related to the immune system and the inflammatory response activation, while the *nfkb2* is part of a complex that activates genes involved in inflammation and immune function, increasing the expression of pro-inflammatory molecules like *tnfα*, *il8*, *il6*, and *il1β*. *Nfkbiaa* is estimated to enable NF-kappa B binding activity, acting upstream of or within negative regulation of hematopoietic progenitor cell differentiation, and *ccl34a* is a chemokine, named Chemokine ligand 34a, essential for the zebrafish immune response [31]. *Hmox1a*, orthologous of the human Heme Oxygenase 1 gene, is necessary for normal development and macrophage migration in zebrafish [32]. Finally, cxcl-cic, orthologous of human CXCL1, is a chemokine that controls neutrophils [33].

The results show that almost all the gene expression levels observed were downregulated after the exposure to both NPs, except for *nf2a* and *il1β*, compared to Ctrl (Figure 4).

### 2.6. SiO_2_-Aerosil200 NPs Significantly Increased the Neutrophil Number

Neutrophils and macrophages work synergically with the immune cells in regulating the immune system to achieve a balance of inflammation and its resolution [34].

Once harmful molecules have been removed, macrophages can eliminate dead neutrophils to return to homeostasis [35]. If this process fails, uncontrolled cytokines will be released, and tissue damage will occur along with possible immune disorders [34]. 

Using Sudan Black b staining, the mature neutrophils were marked in embryos after SiO_2_-RHSK and SiO_2_-Aerosil200 NP exposure and in Ctrl (embryos not exposed).

The embryos were mechanically dechorionated at 24 hpf and exposed to both NPs soon after. In zebrafish embryos, the first immune cell precursors start to be present around the 12 hpf, during the primitive wave of hematopoiesis, giving rise to the macrophages. Approximately at 33 hpf, a further subset of cell precursors differentiates into neutrophils. The definitive hematopoiesis starts around the 24 hpf, when the pluripotent precursors differentiate within the posterior blood island that will expand and become the caudal hematopoietic tissue (CHT). The CHT will give rise to mature macrophages and neutrophils from 2 dpf (days post fertilization) onwards. From 4 dpf, the kidney marrow begins to mature, becoming the site of definitive hematopoiesis in adult zebrafish [36,37].

Considering this, we exposed the embryos to both SiO_2_-RHSK and SiO_2_-Aerosil200 NPs at different concentrations (1, 10, 100, and 200 µg/mL) from the 24 hpf to the 72 hpf (48 h of exposure) and processed the embryos for the histochemical analysis at the end of this period. To evaluate the NPs’ effect, we focused our attention on the CHT region (Figure 5). The results show a significantly increased number of neutrophils in the CHT region of embryos exposed at the highest concentration of SiO_2_-Aerosil200 NPs, compared to embryos exposed to SiO_2_-RHSK NPs and Ctrl embryos. A higher number of neutrophils was also observed in embryos exposed to SiO_2_-RHSK NPs, compared to Ctrl, but this increment was not statistically significant.

## 3. Discussion

In this research work, we focused on the comparative assessment of the biological effects of silica rice husk (SiO_2_-RHSK), nanoparticles composed of amorphous silica extracted from rice husk, a very recalcitrant waste from rice mills, the open combustion of which, operated for preventing its accumulation on the soil, can result in the generation of carcinogenic gases, and the commercial fumed silica hydrophilic (SiO_2_-Aerosil200) nanoparticles.

After the NPs’ characterization, using various methods such as Transmission Electron Microscopy (TEM) and Dynamic Light Scattering (DLS), we evaluated the NPs’ toxicity on zebrafish embryos by the FET test (OECD No. 236, 2013).

We observed a very low mortality rate and negligible sub-lethal defects in embryos exposed to both SiO_2_-RHSK and SiO_2_-Aerosil200 NPs, compared to Ctrl. These data underlined that both SiO_2_ NPs do not show acute toxicity during zebrafish development, probably due to the presence of chorion, which prevents direct interaction with embryos [38]. To date, no effects were observed on the chorion and the hatching activity, contrary to what happens after exposure to many other NMs, such as CuO, ZnO, and Ag NPs [39,40], testifying to the safety profile of these SiO_2_-based NM under acute exposure conditions.

Our results are in accordance with what was reported in the literature by Hong and colleagues [27], in which the toxicity of silica nanoparticles recovered and purified from rice husk (RH) through thermochemical processes was evaluated. Hong and collaborators used 200 μg/mL as a higher concentration, without observing any significant differences in survival and hatching of exposed embryos compared to controls.

At this point, we decided to evaluate the possible neurotoxic effects of SiO_2_-RHSK and SiO_2_-Aerosil200 NPs on *Tg(-3.1neurog1:GFP)sb2* embryos chorion deprived by evaluating *neurogenin1* expression and by molecular analyses of some the major genes, such as *rpl13am*, *β-synuclein*, *parkin*, *pink1*, *uch-l1*, *park7*, *atg5*, *ambra1a*, *gfap*, and *gap43*, known to be related to silica-induced neurotoxicity [41].

The *Tg(-3.1neurog1:GFP)sb2* is a transgenic zebrafish line characterized by the expression of GFP under the promoter of proneural factor neurogenin1 [26,42], which is a key regulator of dorsal root ganglion (DRG) neuron development [43]. The purpose was to observe if SiO_2_-RHSK and SiO_2_-Aerosil200 NP exposure could interact with this gene, possibly observing a GFP reduction as a readout [44]. The results obtained on transgenic embryos do not show any differences in GFP fluorescent signal compared to Ctrl, meaning that both NPs do not directly affect the *neurogenin1* pattern or do not pass through the epidermis.

The gene expression evaluation showed that both NPs do not have significant effects on most of the mRNA levels analyzed, except for significantly increased levels of *gap43*, present in the neuronal growth cones during synaptogenesis and axonal development/regeneration [29].

We hypothesize that this result, obtained in embryos exposed to both NPs, is related to the beneficial effect that SiO_2_-RHSK and SiO_2_-Aerosil200 NPs have on synaptogenesis. Following the results reported by Fan and colleagues, the levels of *gap43* mRNA are elevated between day 2 and day 5, since synaptogenesis is known to be a process that occurs during later stages of nervous system formation, consequently showing a peak of *gap43* transcript expression levels at later stages of development [45].

The significantly increased *gap43* levels that we detected in embryos at 5 dpf after SiO_2_-RHSK and SiO_2_-Aerosil200 NP exposure were not observed by Fan and his team after the ethanol treatment, a known neurotoxicant substance, confirming that both NPs are not causing neurotoxicity in our experiment. Furthermore, this result is corroborated by the data obtained by Bei and colleagues, who noted that neuronal marker levels are increased by silica NPs, which also showed the capacity to increment maturity in human neural stem cells (hNSCs) [46].

To deeply evaluate if both SiO_2_-RHSK and SiO_2_-Aerosil200 NPs could be considered beneficial and safe NPs, we decided to assess their eventual pro-inflammatory capacities by observing expression levels of *nrf2a*, *nrf2b*, *tnfα*, *nfkb2*, *il8*, *il6*, *il1β*, *nfkbiaa*, *ccl34a.4*, *hmox1a,* and *cxcl-cic* genes.

The qRT-PCR results showed that most of the gene expression levels observed were downregulated after exposure to both NPs, except for *nfr2a* and *il1β*, compared to Ctrl. The only significant result was the downregulation of *ccl34a.4* gene levels in embryos exposed to SiO_2_-Aerosil200 NPs, compared to Ctrl embryos and embryos exposed to SiO_2_-RHSK NPs.

Recently, the *ccl34a.4* (chemokine ligand 34a.4) gene was related to a specific macro-phage subpopulation, the ccl34a.4+ pro-remodeling macrophages, present in the barrier tissues and internal organs. The most prominent feature of the ccl34a.4+-macrophages is the extensive expression of genes related to tissue remodeling and regeneration that they perform, including those associated with immune regulation, such as *ccl34a.4* [47].

The ccl34a.4+-macrophages are a subpopulation of the tissue-resident macrophages (TRMs) that, alongside dendritic cells (DCs) are fundamental components of the innate immune system, playing essential roles in tissue homeostasis and repair, in addition to tissue development and immunity [48,49]. The tissue-resident macrophages were initially divided into two populations, the pro-inflammatory M1 macrophages and the anti-inflammatory M2 macrophages, presenting tissue remodeling functions that depend on the external stimuli signals. Successively, thanks to single-cell RNA sequencing and fate-mapping analyses, it was possible to reveal the presence of distinct TRM subsets coexisting across organs [47].

The result that we observed, which is the significant downregulation of *ccl34a.4* mRNA levels in embryos after SiO_2_-Aerosil200 NP exposure, led us to hypothesize that these commercial NPs may interfere with the correct function of macrophages which, along with neutrophils, constitute a fundamental defense during inflammation. This process, which consists of recognition, removal of harmful objects, and repair of tissues, is the host protection response to infections and tissue injury, including a tightly regulated program essential for maintaining homeostasis [34].

Since we observed a possible deregulation of macrophages after SiO_2_-Aerosil200 NP exposure, alongside an increased trend in *il1β* expression levels in embryos exposed to SiO_2_-RHSK NPs, but not to SiO_2_-Aerosil200 NPs, we decided to investigate the neutrophil number at the histological level. To date, *il1β* is related to neutrophil recruitment through the expression of adhesion molecules on the endothelium, and the production of local chemokines [50].

The results showed a significantly increased number of neutrophils in the CHT region of embryos exposed to SiO_2_-Aerosil200 NPs, compared to embryos exposed to SiO_2_-RHSK NPs and Ctrl. An increasing trend in neutrophil number is visible in embryos exposed to SiO_2_-RHSK NPs compared to Ctrl, even if not statistically significant. We hypothesize that this increment is normal, considering the exposition to NPs, which will be recognized by host defense as endogenous agents and thus potentially dangerous.

The evidence that the SiO_2_-Aerosil200 NPs are recognized by the immune defense more markedly than SiO_2_-RHSK NPs lets us make some hypotheses.

The first hypothesis is related to the size of SiO_2_-Aerosil200 NP aggregates. Based on the DLS analysis, the SiO_2_-Aerosil200 NPs aggregates were smaller than the ones formed by SiO_2_-RHSK NPs. This suggests a higher probability of uptake through the embryos’ epidermis and a consequent easier recognition by the innate immune system. For instance, the hydrodynamic diameter detected for SiO_2_-Aerosil200 NPs was 180.13 ± 0.92 in milliQ and 177.63 ± 1.42 in FET, while for SiO_2_-RHSK NPs was 538.87 ± 126.88 in milliQ and 846.97 ± 12.24 in FET. 

The second hypothesis is associated with the effect that SiO_2_-Aerosil200 NPs exert on macrophages. Macrophages and neutrophils work together with adaptive immune cells to regulate the immune system, achieving a balance of inflammation and resolution. Neutrophils can be eliminated by macrophages, to return to homeostasis once dangerous molecules are removed [34]. Contemporarily, the apoptotic neutrophils operate to switch the macrophage profile from pro- to anti-inflammatory (M1 to M2), suppressing inflammatory cytokines such as TNF and IL8 [51].

The downregulation of ccl34a.4 macrophages that we observed at the molecular level could be related to dysfunction in this type of cells, interfering with the production of anti-inflammatory macrophages. This would lead to an imbalance of inflammation and its resolution, as well as an uncontrolled increase in the number of neutrophils, which are trying to sedate the inflammatory response.

For these reasons, we hypothesize that the significant increase in neutrophil number observed in embryos exposed to SiO_2_-Aerosil200 NPs is probably related to the detrimental effect that these NPs have on macrophages.

The SiO_2_-RHSK NPs, instead, do not show any significant adverse effect on macrophages or other actors of the innate immune system.

Although many studies are devoted to deriving SiO_2_ from bio-sources, to date, the knowledge on their biocompatibility profile is still underexplored. Solarska-Sciuk and colleagues [52] recently reported that the bio-silica extracted from *Urtica dioica* is more hazardous than the pyrogenic one in in vitro systems, pointing out the necessity to carefully assess the safety of the naturally derived SiO_2_ NPs. Thus, also considering the vast variety of SiO_2_ nanoforms available in the future by using the different synthesis techniques and sources, more efforts in the research aimed at characterizing their biological interactions are desirable.

## 4. Materials and Methods

### 4.1. Animal Rearing and Ethics

AB line and transgenic line *Tg(-3.1neurog1:GFP)sb2* were used in this work. Adults were raised with constant temperature (28.5 °C) and regular photo rhythm (light for 14 h/dark for 10 h). Fish were fed with dry food (Zebrafeed by Sparos, Olhão, Portugal) twice a day. The circulating water system was kept with controlled conductivity and pH. Experimental operations related to zebrafish were pursued following the guidance issued by the Animal Ethical and Welfare Committee at the University of Milano-Bicocca.

### 4.2. SiO_2_-RHSK NPs Production

Bio-silica NPs (SiO_2_-RHSK NPs) were produced from rice hush (RH) after acidic digestion and calcination process by CENTITVC (Centre for Nanotechnology and Smart Materials, Vila Nova de Famelicao, Portugal) in the framework of the H2020 BIOMAT project. Briefly, rice husk residues were crushed to a desirable size (around 1.5 mm) and washed with distilled water for 2 h at room temperature. Successively, HCl 10% was added to the previous filtrated RH and remained in contact with it for another 2 h at 60 °C. Finally, the neutralized RH was dried overnight, milled at 0.5 mm, and calcinated at 22 °C to 700 °C for 2 h and 30 min, and at 700 °C for 3 h and 30 min.

### 4.3. SiO_2_-Aerosil200 Production

The hydrophilic fumed NPs (d = 12 nm, SiO_2_ 99.8 wt%, AEROSIL^®^ 200, Evonik Industries) were produced by Degussa, a German Chemical industry in Essen, Germany. The production process is called pyrogenetic and is known as “Aerosil”. It requires very high temperatures (around 1000 and 2500 °C) and consists of the flame hydrolysis of SiCl_4_ (silicon tetrachloride).

### 4.4. Characterization of SiO_2_-Aerosil200 and SiO_2_-RHSK NPs

Both SiO_2_-RHSK and SiO_2_-Aerosil200 NPs were weighed and resuspended in milliQ water to a final concentration of 1 mg/mL. Successively, the NPs were sonicated for 10 min with a probe sonicator (probe MS72, 40 W, pulse 1.00/1.00, no-stop, 10 min; Bandelin^®^, GmbH & Co. KG, Berlin, Germany), using ice to avoid overheating.

The morphology and size of SiO_2_-RHSK and SiO_2_-Aerosil200 NPs were scanned at 200 kV by TEM (JEM-2100; JEOL Ltd., Tokyo, Japan) using a Rio Complementary Metal-Oxide-Superconductor (CMOS) equipped with an 8-megapixel Gatan camera (Gatan, Warrendale, PA, USA). Then, the mean size, standard deviation (SD), and size distribution of NPs were measured by Fiji (ImageJ) [53]. NP solutions were prepared by adding milliQ water to obtain a final concentration of 100 µg/mL.

The hydrodynamic sizes, distribution, and ζ-potentials were determined by dynamic light scattering (DLS) using Zetasizer^®^ nano-zs90 (Malvern Instruments, Malvern, UK). Specifically, to measure the ζ-average and the poly dispersion index (PdI) were used the DTS0012 cuvettes, while to measure the ζ-potential were used the DTS1070 cuvettes. 

FTIR-ATR analysis of both SiO_2_-RHSK NPs and SiO_2_-Aerosil200 particles was performed using a Perkin Elmer Spectrum 100 Series (PerkinElmer Life and Analytical Sciences, Shelton, WA, USA) spectrophotometer with a spectral range and resolution factor of 4000 to 650 cm^−^^1^ and 8 cm^−^^1^, respectively.

### 4.5. Fish Embryo Toxicity (FET) Test and SiO_2_-RHSK and SiO_2_-Aerosil200 NPs Exposure

The FET test was performed following the OECD guidelines n. 236 (2013), as described in Bragato et al., 2022 [54].

Briefly, after fertilization, embryos from the zebrafish AB line were exposed to NPs in a 24-multiwell plate at the concentrations of 0.1, 1, 10, and 100 μg/mL. For this project, the total embryos observed were n = 120 (SiO_2_-RHSK NPs) and n = 120 (SiO_2_-Aerosil200 NPs) in 5 different experiments.

Every 24 h, embryos were screened for lethality, in particular checking for coagulation of fertilized eggs, lack of somite formation, lack of detachment of the tail bud from the yolk sac, and lack of heartbeat, according to the specific time points.

Moreover, sub-lethal endpoints, such as reduced yolk resorption, blood congestion, formation of edema, and lack of hatch, were observed from the 48 to 96 hpf developmental stage.

Sonication was performed before the NPs were added into FET solution (0.1 g of NaHCO_3_; 0.1 g of Instant Ocean; 0.19 g of CaSO_4_·2H_2_O for 1 L of solution).

### 4.6. Morphometric Analysis of Developing Zebrafish Embryos

The morphometric analysis of developing zebrafish embryos was performed to study developmental delays or allometric growth alterations of embryonic structures, both being important indicators of toxicity [55].

At the end of the FET test, the embryos were fixed in 4% methanol-free paraformaldehyde for 2 h at room temperature (RT) or overnight (O/N) at 4 °C and rinsed several times in PBS washing buffer. Successively, the embryos were photographed under a Zeiss stereomicroscope (Stemi SV6) equipped with a digital camera (AxioCam ERc5s) (Carl Zeiss S.p.A., Milan, Italy), positioning the embryos laterally (left side) and dorsally using methylcellulose 3%. The images were then processed by the Zen 3.4 software (blue edition) to quantify the body length, eye distance, eye area and diameter, and yolk area.

### 4.7. Dechorionation End Exposure to NPs

To remove chorion from zebrafish embryos, two approaches are commonly used, such as manual removal with forceps and enzymatic degradation with pronase treatment [56]. In this research, the chorion was removed by pinching the chorion surface and pulling outward to tear the chorion open and releasing the embryos at 24 hpf.

Soon after the chorion removal, the embryos of transgenic line *Tg(-3.1neurog1:GFP)sb2* were exposed to SiO_2_-RHSK and SiO_2_-Aerosil200 NPs at a concentration of 0.1, 1, 10, and 100 μg/mL (successively also 200 μg/mL) until the 96 hpf developmental stage.

In this case, 10 embryos were exposed in a 60 mm Petri dish, using 10 mL of FET solution. The experiment was repeated three times (total embryos used in three experiments n = 130 for each NP).

### 4.8. RNA Extraction and Reverse Transcription

Total RNA was extracted from zebrafish embryos at 96 hpf using TRI Reagent (MRC, Cincinnati, OH, USA). First-strand cDNA synthesis reaction from total RNA was catalyzed by LunaScript^®^ RT SuperMix Kit (NEB #E3010, New England Biolabs, Ipswich, MA, USA). The cDNA was then amplified with Actin-β primers (Fw: 5′-TGTTTTCCCCTCCATTGTTGG-3′, Rw: 5′-TTCTCCTTGATGTCACGGAC-3′) using Phusion High-Fidelity polymerase (Finnzymes, Thermo Fisher Scientific, Waltham, MA, USA).

### 4.9. Real-Time Quantitative PCR

Quantitative real-time PCR (qRT-PCR) was performed as reported in Bragato et al., 2024 (submitted) [31]. Briefly, the Luna^®^ Universal qPCR Master Mix kit (New England Biolabs, Ipswich, MA, USA) and the Quantum 3™ (Applied Biosystems, Waltham, MA, USA) Real-Time PCR system were used according to manufacturer instructions. Real-time PCR was performed in a 10 μL reaction containing 600 nM of each primer, 2 μL template cDNA, and 5 μL qPCR Master Mix. The PCR was run at 95 °C for 60 s followed by 40 cycles of 95 °C for 15 s and 60 °C for 30 s. lsm12b or mobk13 were used as the endogenous control. Relative changes in gene expression between control and treated samples were determined using the 2^−ΔΔCt^ method [57], and the results were presented in logarithmic scale fold change [28]. The primer sequences of tested genes (*nrf2a*, *nrf2b*, *tnfα*, *nfkb2*, *il8*, *il6*, *il1β*, *nfkbiaa*, *ccl34a.4*, *hmox1a*, *cxcl-cic*, *β-syn*, *park7*, *rpfl13a*, *atg5*, *gfap*, *ambra1a*, *ulk1a*, *uch-1l*, *gap43*, *parkin*, *pink1,* and *beclin1*) are listed in Appendix A.

### 4.10. Sudan Black Staining

To mark the mature neutrophils, we used Sudan Black (SB) histochemical staining, as previously described in Bragato et al. 2024 [31]. Briefly, the embryos were fixed with 4% methanol-free paraformaldehyde (PFA) in PBS for 2 h at RT, rinsed in PBS several times, incubated in SB for 20 min at RT, washed extensively in 70% ethanol in water, then progressively rehydrated with PBS plus 0.1% Tween-20 (PBST).

### 4.11. Neutrophils Quantification

To quantify the Sudan Black staining, embryos were observed under a stereo microscope, and pictures of the caudal hematopoietic tissue (CHT) area were collected. The purple signal area (expressed as arbitrary units) was calculated by ImageJ Fiji software (https://imagej.net/Fiji, accessed on 8 February 2023) on bright-field images taken under the same conditions, such as light exposition and frame size, as reported in Bragato et al., 2024 [31]. 

### 4.12. Statistical Analyses

Data are reported as means and standard error of the means (SEM) using GraphPad Prism 9.2.0.332 (GraphPad Software). The statistical significance between multiple groups was determined by one-way ANOVA, followed by Bonferroni’s post hoc test analysis. The level of significance was considered at *p* < 0.05.

## 5. Conclusions

In summary, our results showed that the exposure to SiO_2_-RHSK and SiO_2_-Aerosil200 NPs caused negligible embryotoxicity, but no neurotoxicity or inflammation in zebrafish embryos at the concentrations used (0.1, 1, 10, 100, and 200 µg/mL). Interestingly, the increased *gap43* expression levels suggested a possible beneficial role of both SiO_2_-RHSK and SiO_2_-Aerosil200 NPs on neural stem cells, deserving attention and further evaluations.

However, SiO_2_-Aerosil200 NPs were demonstrated to have slight immunomodulatory activity, affecting the macrophages, in particular the *ccl34a.4+* macrophage subpopulation.

To conclude, the bio-based SiO_2_-RHSK NPs were revealed to be safer than SiO_2_-Aerosil200 NPs, even if there will be the necessity to identify concentration–response relationships for subacute and chronic effects.

Nonetheless, the new bio-nanomaterials demonstrate great potential for developing new safe and sustainable nanotechnologies in different industrial sectors in the future.

## Figures and Tables

**Figure 1 ijms-25-05530-f001:**
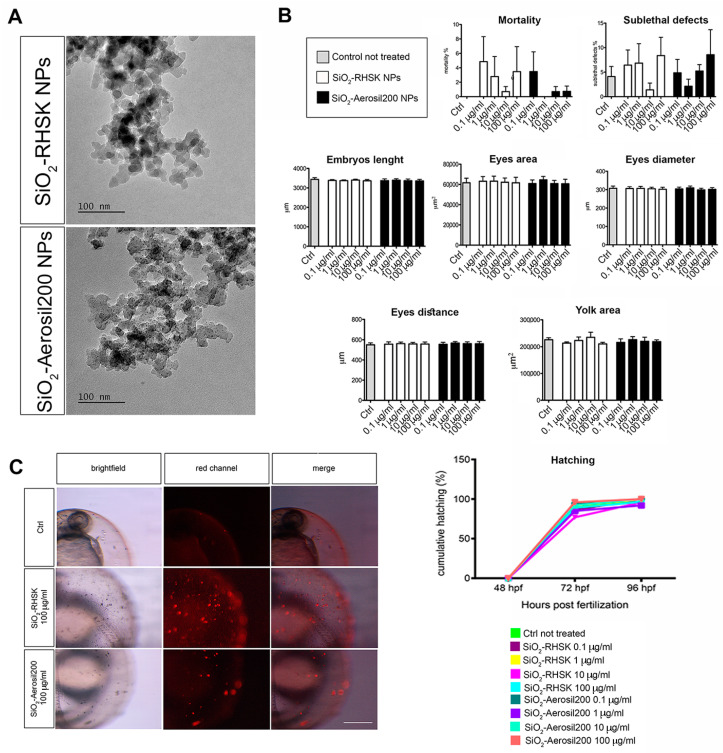
SiO_2_-RHSK and SiO_2_-Aerosil200 NPs and their interaction with embryos. (**A**) TEM images of SiO_2_-RHSK and SiO_2_-Aerosil200 NPs; (**B**) mortality and sub-lethal defects reported as a percentage, measures (embryo body length, eye area, eye diameter, eye distance, and yolk area) and hatching analyses performed on embryos at 96 h post fertilization (hpf) after exposure to both SiO_2_-RHSK and SiO_2_-Aerosil200 NPs at different concentrations; (**C**) representative images of NP agglomerates on the chorion of embryos (red signal) exposed to both NPs at 100 µg/mL. Scale bar = 100 µm.

**Figure 2 ijms-25-05530-f002:**
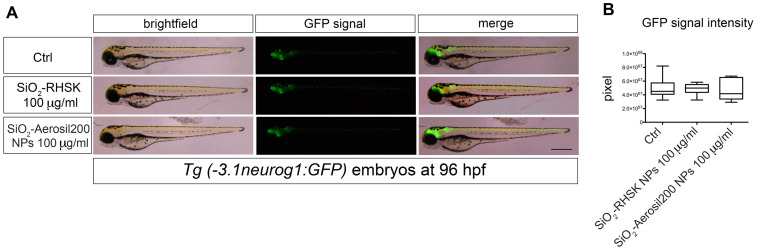
Neurogenin1 signal is not compromised by the exposure to SiO_2_-RHSK and SiO_2_-Aerosil200 NPs. (**A**) Representative images of *Tg(-3.1neurog1:GFP)sb2* embryos at 96 hpf exposed to NPs, compared to Ctrl. The green signal correspond to GFP. (**B**) Quantitation of the GFP green signal in embryos exposed to both SiO_2_-RHSK and SiO_2_-Aerosil200 NPs and in Ctrl (embryos not exposed). Results were achieved from 3 independent experiments each using 20 embryos exposed to SiO_2_-RHSK NPs, 20 embryos exposed to SiO_2_-Aerosil200 NPs, and 20 embryos not exposed (Ctrl). Scale bar = 100 μm.

**Figure 3 ijms-25-05530-f003:**
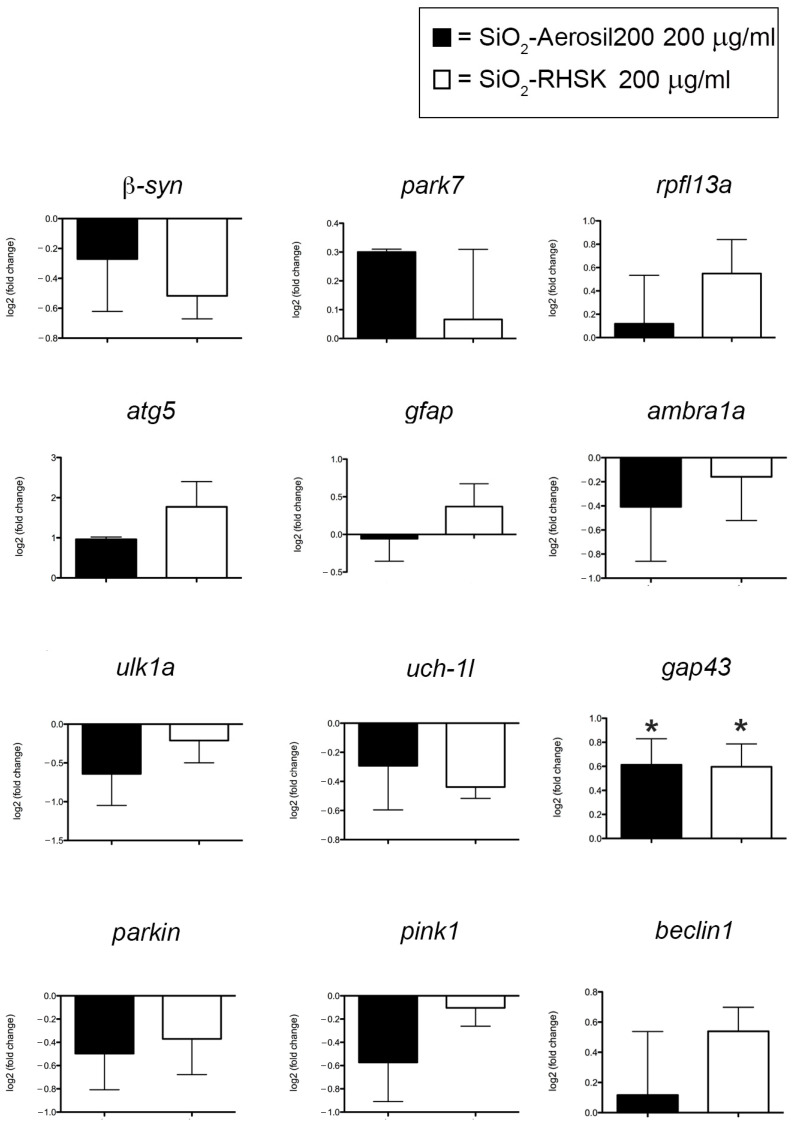
Expression levels of genes related to neurotoxicity caused by the exposure to SiO_2_, in embryos at 96 hpf. Results were achieved from 3 independent experiments each using cDNA obtained after RNA extraction from 20 embryos pulled together. * *p* < 0.5 with respect to Ctrl.

**Figure 4 ijms-25-05530-f004:**
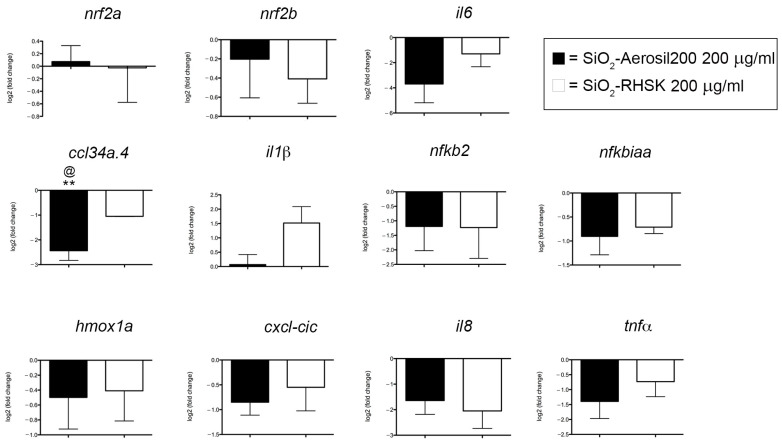
Expression levels of genes related to inflammation and oxidative stress, in embryos at 96 hpf. Results were achieved from 3 independent experiments each using cDNA obtained after RNA extraction from 20 embryos pulled together. ** *p* < 0.01 with respect to Ctrl; ^@^
*p* < 0.5 with respect to SiO_2_-RHSK 200 µg/mL.

**Figure 5 ijms-25-05530-f005:**
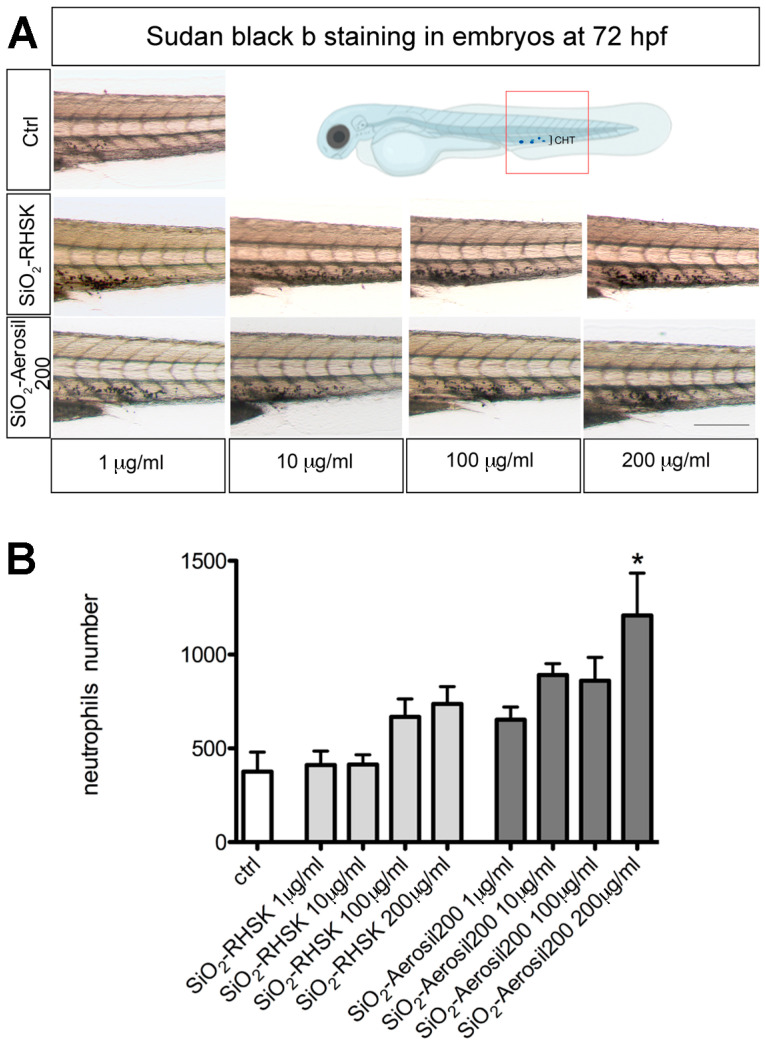
Sudan Black b staining on embryos after NP exposure. (**A**) Representative images of embryos showing the neutrophils in the caudal hematopoietic tissue (CHT) region (highlighted by the red box in the illustration). The CHT was observed and analyzed using Imagej (Fiji) software (2.9.0/14 September 2022). (**B**) Data represent the mean ± SEM of 3 independent experiments (n = 60 embryos for each condition). Scale bar: 100 µm. * *p* < 0.5 with respect to ctrl.

**Table 1 ijms-25-05530-t001:** Dynamic Light Scattering (DLS) and Transmission Electron Microscopy analyses. Hydrodynamic diameter (d.nm), polydispersion index (PdI) ζ-potential (mV), TEM esteemed diameter (nm), and NPs’ shape of 1 mg/L SiO_2_-RHSK and SiO_2_-Areosil200 NPs measured by dynamic light scattering (DLS) in different media (mQ water and FET medium). Means SD of three replicates.

Nanoparticles (NPs)	Hydrodynamic Diameter (d.nm)	Polydispersion Index (Pdl)	ζ-Potential (mV)	TEM Esteemed Diameter (nm)	NPs Shape (TEM)
**SiO** **_2_-RHSK**	H_2_O milliQ: 538.87 ± 126.88	H_2_O milliQ: 0.65 ± 0.13	H_2_O milliQ: −22.80 ± 0.14	<30	Polyhedral
	FET medium: 846.97 ± 12.24	FET medium: 0.93 ± 0.05			
**SiO** **_2_-Aerosil200**	H_2_O milliQ: 180.13 ± 0.92	H_2_O milliQ: 0.13 ± 0.01	H_2_O milliQ: −21.03 ± 0.62	<30	Polyhedral
	FET medium: 177.63 ± 1.42	FET medium: 0.12 ± 0.04			

## Data Availability

The data that support the findings of this study are available from the corresponding author upon reasonable request.

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
