# Peer review of "Biocompatibility Analysis of Bio-Based and Synthetic Silica Nanoparticles during Early Zebrafish Development"

_ijms, 2024, doi:10.3390/ijms25105530_

Round 1
Reviewer 1 Report
Comments and Suggestions for Authors
The study seems interesting and results are novel. However, major revision is needed.
Specific comments:
The quality of figures is very low. Figures are very hard to read. This issue must be improved.
The manuscript should be re-written as some parts are very hard to read and/or too long, e.g., introduction section is too long. The title is not informative and confusing: “Comparative assessment of bio-based and synthetic silica nanoparticles during early zebrafish development”. Maybe “Safety assessment of bio-based and synthetic silica nanoparticles during early zebrafish development” or “Analysis of biocompatibility of bio-based and synthetic silica nanoparticles during early zebrafish development”.
The authors have considered to study changes at mRNA levels of selected markers of neurotoxicity, inflammation, and oxidative stress. These markers should be also studied at protein levels. Please note that only changes at protein levels (or in this case, no changes due to nanomaterial biocompatibility) may have biological importance. Please address this issue.
Comments on the Quality of English LanguageSome parts require English language correction (moderate changes are needed).
Reviewer 2 Report
Comments and Suggestions for Authors
This manuscript deals with the effects of amorphous silica bio-based silica nanoparticles from rice husk and commercial hydrophilic fumed silica nanoparticles on early zebrafish development. This manuscript nicely demonstrated biological effects and safety of these nanoparticles, which would attract even general non-specialist readers. From this positive view point, I may recommend publication of this work in Int. J. Mol. Sci. However, several data had better be added upon revisions. Please see below.
1) Surface nature (functional groups) are important keys for interaction with biomatters and cells. Therefore, appropriate characterization methods such as IR spectroscopy had better be applied to both the silica nanoparticles. Judging from similarity in zeta potentials, so obvious differences between two particles will not be found. experimental confirmation is important.
2) Ther are huge varieties of artificially prepared silica nanoparticles and bio-originated silica nanoparticles. Therefore, these two examples cannot represent artificial and biological nanoparticles. Please well consider this logic. Hopefully, several other examples of other silica nanoparticles are tested.
3) In this manuscript, there are two short paragraphs (one sentence paragraph) are included. Therefore, organization of text description does not look good. Please avoid use of too short paragraphs.
Round 2
Reviewer 1 Report
Comments and Suggestions for Authors
The manuscript was partially improved according to my comments. The paper can be now accepted for publication.
Reviewer 2 Report
Comments and Suggestions for Authors
Replies and revisions are fine. The revised version becomes acceptable.